# Emotion Regulation with Parents and Friends and Adolescent Internalizing and Externalizing Behavior

**DOI:** 10.3390/children8040299

**Published:** 2021-04-13

**Authors:** Eric W. Lindsey

**Affiliations:** Psychology Department Berks Campus, Penn State University, Reading, PA 19610, USA; EWL10@psu.edu

**Keywords:** parent–adolescent relationship, friendship, emotion regulation, internalizing behavior, externalizing behavior

## Abstract

This study examined adolescents’ self-reported use of emotion regulation strategies with parents and friends in relation to internalizing and externalizing behavior. A total of 185 children aged 13–14 years old (91 girls, 94 boys) completed three surveys to assess their emotion regulation strategies with mothers, fathers and best friends. Parents completed surveys assessing adolescents’ internalizing and externalizing behavior. Regression analysis revealed that adolescents’ self-reported ER with mothers and fathers and friends made independent contributions to parent reports of youth internalizing and externalizing behavior. Adolescents who reported engaging in more emotion suppression with friends had higher internalizing scores, whereas adolescents who reported more affective expression with friends had lower internalizing scores. Self-reported emotion regulation strategies with mothers and fathers were unrelated to internalizing behavior. Adolescents who reported engaging in higher levels of affective suppression and cognitive reappraisal with their mothers and fathers had lower parental ratings of externalizing behavior. Emotion regulation strategies with best friends were unrelated to externalizing behavior.

## 1. Introduction

The transition to adolescence is characterized by elevated risk for almost all mental health disorders [1]. Psychopathology symptoms during adolescence generally emerge in two broad domains: (1) internalizing disorders, characterized by symptoms that include emotional, mood, and somatic disturbances; and (2) externalizing disorders, characterized by symptoms of developmentally elevated levels of aggression, substance use, and delinquent behavior [2,3,4]. Rates of psychiatric hospitalization among adolescents in the United States have increased over the past 20 years [5,6,7] for both internalizing and externalizing disorders. However, internalizing and externalizing disorders have different developmental trajectories. For example, the prevalence of depressive symptoms increases dramatically across the transition to adolescence and peaks during middle adolescence [8]. In contrast, externalizing symptoms, such as conduct disorder and delinquency, increase steadily across the adolescent years to peak in young adulthood. At the same time, there is clear evidence of comorbidity across internalizing and externalizing symptoms of psychopathology, as well as between subdomains of internalizing and externalizing behaviors [9].

Theoretical and empirical evidence suggest that there may be common underlying disruption to normative domains of functioning linked to multiple psychopathology symptom dimensions and disorders. One functional domain identified by researchers as being central to psychopathology is emotion regulation [10,11]. Emotion regulation refers to the assorted processes by which emotions are altered to meet situational demands. Most definitions of emotion regulation focus on emotion regulation in the self, or the methods individuals use to specify the expression of discrete emotions, control the timing of emotion, and influence experience and expression of emotion. In addition, emotion regulation is viewed as involving changes in experiential, behavioral, and physiological domains. Emotion regulation has been identified as a key component in the trajectory of internalizing [12] and externalizing problems [13] during adolescence. Specifically, in a comprehensive quantitative meta-analysis of research examining associations between emotion regulation and internalizing symptoms, Schäfer and colleagues [14] found significant medium negative associations between emotion regulation and assessments of depression and anxiety in adolescents. Around the same time Compas et al. [13] conducted a meta-analysis of research (i.e., 212 studies with 80,850 participants) on the association between emotion regulation and symptoms of internalizing and externalizing behavior in childhood and adolescence. The authors found that emotion regulation, broadly defined, had a significant medium negative association with both internalizing and externalizing symptoms in cross-sectional studies. A major conclusion of both meta-analyses [13,14] was the need for additional research to provide a more detailed account of how specific emotion regulation strategies may be linked to internalizing and externalizing symptoms. The authors of both meta-analyses also called for more research on the role that the context in which emotion regulation occurs plays in connections between emotion regulation strategies and internalizing and externalizing symptoms.

The Extended Process Model (EPM) has received empirical support for its outline of different emotion regulation strategies [15]. The EPM defines emotion regulation as the way a person responds to mental and environmental stimuli, to maintain adaptive, goal-oriented functioning [16]. Multiple families of emotion regulation processes are identified in the EPM, organized on a continuum from relatively distal to more proximal strategies in relation to their primary impact on the emotion-generative process. Three of the more proximal emotion regulation strategies identified by the EPM that have been linked to adolescents’ internalizing and externalizing behaviors include (a) cognitive reappraisal, (b) affective suppression and (c) affective expression. Cognitive change or reappraisal is defined as an active mental process of interpreting the meaning of a situation in an attempt to influence one’s emotions [17]. Typically, reappraisal comprises interpreting a potentially emotion-eliciting situation in nonemotional terms or in a way that changes its emotional impact. In contrast, affective suppression is defined as a response-focused strategy that occurs somewhat late in the emotion-generative process, and principally alters the behavioral component of the emotion response tendencies. The behavioral focus of suppression also means that it is a strategy that does not reduce the experience of negative emotion, because the emotion is not specifically targeted and may thus persist and accrue unredressed. Affective expression, also referred to as affective acceptance or expressiveness engagement, is another emotion regulation strategy designed to directly regulate emotion by engaging expressive dynamics in order to moderate the emotional experience.

Cognitive reappraisal is considered to be a more optimal emotion regulation strategy for psychosocial adjustment compared to expressive suppression [12,18,19]. In contexts that call for the downregulation of negative emotion, reappraisal helps to reduce the experiential and behavioral components of negative emotion. Individuals who regularly use reappraisal putatively feel and communicate more positive, and less negative, emotion, and have higher levels of personal well-being. Elevated experience of positive emotion is considered to play a protective role in susceptibility to depression and anxiety, and to reduce the enactment of externalizing behavior problems. By contrast, the emotion regulation strategy of suppression, which according to the EPM arises later in the emotion formation sequence, necessitates that the individual coordinate emotion response tendencies as they recurrently arise. These recurrent exertions may expend cognitive resources that could otherwise be used for successfully navigating social interactions in which the emotions arise. Suppression is also considered to be detrimental to well-being because it creates in the individual a sense of incongruence, or discrepancy, between their inner experience and outer expression [20], leading to negative self-perceptions and feelings of being inauthentic in one’s social relationships [21]. Such negative self-perceptions may contribute to internalizing problems such as depression and anxiety, and to externalizing problems in the form of acting out behaviors. Affective expression has received very little attention in empirical work, thus it is unknown how it may relate to the manifestation of internalizing and externalizing problems.

Studies examining adolescents’ use of emotion regulation strategies identified by the EPM are notably limited [22], but suggest that youth use cognitive regulation strategies, including reappraisal, less frequently than adults [23]. Even so, adolescents’ use of reappraisal is more adaptive than other emotion regulation strategies [24,25]. Specifically, evidence supports the precepts of the EPM in that use of cognitive reappraisal by adolescents is associated with less internalizing behavior, whereas emotional suppression is associated with more internalizing symptoms [13,14]. A notable gap in the literature, however, is the lack of investigation of links between emotion regulation strategies and externalizing problems [14].

A key tenant of the EPM model is that emotion regulation strategies must be considered as part of a person–situation interaction [26]. Any given interaction has a particular meaning to the individual and that meaning forms the foundation for a multisystem response that includes the experience and regulation of emotion [27]. The identity of a social partner is recognized as an important element of the person–situation transaction [28]. The EPM suggests that different social relationships can have unique meaning to an individual, so that emotions experienced with one partner can be interpreted differently from emotions experienced with another partner. In turn, the ER strategy used in reference to an emotion can vary from one relationship to the next, with the result being that, for any given person, emotion regulation may look very different in one relationship compared to another. Such a proposition is consistent with theories that suggest relationship goals may play a role in differences in children’s behavior with different social partners [29,30]. To date, this proposition has yet to be examined with different relationship contexts in adolescence. In the present study, relationships with parents and friends are examined as two distinct contexts in which adolescents may use different emotion regulation strategies.

### 1.1. Parent–Adolescent Relationships

Parent–adolescent relationships have a hierarchical structure founded on parental authority, and parental provision of resources and support to the adolescent [31]. Parent–adolescent relationships are also characterized by an inherent stability, built on a history of interactional experiences across multiple situations. In the context of more stable relationships with parents where adolescents are working to establish autonomy [32], youth may use emotion regulation strategies that focus on long-term relationship goals and future emotional displays. With parents adolescents may have developed particular patterns of appraising their emotional reactions so that there is less need to engage in reappraisal as an emotion regulation strategy. Likewise, evidence suggests that in interactions with parents, adolescents use emotion regulation strategies that focus on suppressing emotion [33]. The adolescent years are characterized by a decrease in emotional expression between parents and children [32,33], suggesting that adolescents may be less likely to use affective expression as an emotion regulation strategy when interacting with parents. Although adolescents’ use of different emotion regulation strategies with parents have not been examined, studies have found that individual differences in adolescents’ use of emotion regulation in the context of the parent–child relationship is related to adolescents’ psychosocial adjustment. For example, Yap et al. [34] reported that adolescents who had poor emotion regulation and whose mother’s had higher levels of negative emotional expressivity reported more depressive symptoms. Similarly, Buckholdt et al. [35] found that adolescents who had difficulty regulating their emotions and whose mother’s demonstrated invalidating responses to adolescents’ negative emotions were a greater risk for internalizing and externalizing symptoms.

A contextual view raises the possibility that adolescents use different emotion regulation strategies in the context of their relationship with their mothers and fathers. Indirect support for this possibility comes from empirical evidence of emotion communication differences in mother—child and father—child interactions, with mothers being more emotionally communicative and more encouraging and accepting of adolescents’ expression of emotion than fathers [36,37]. Stocker et al. [38] suggest that mothers’ greater emotional availability may lead adolescents to feel more comfortable expressing emotions during interactions with mothers than with fathers. When asked about their responses to children’s emotion expressions, mothers reported being more likely to encourage sadness expressions and respond constructively to the expression of sadness, whereas fathers reported being more likely to encourage inhibition of sadness [39]. In this way, mother–child and father–child relationships may provide children with different emotion regulation experiences. There is also evidence to suggest that adolescents’ gender accounts for differences in how parents socialize adolescents’ emotional expressivity [40]. Parents talk about emotions more with daughters than sons, which may lead girls to feel more comfortable expressing emotion and to rely less on suppression as an emotion regulation strategy with parents [41]. To date, to the best of my knowledge, no study has examined differences in adolescents’ emotion regulation strategy use with mothers and fathers.

### 1.2. Adolescent Friendships

From middle childhood into adolescence, youth increasingly turn to friends for emotional support and to explore their identity outside their family [42,43,44]. Likewise, with the transition to adolescence, children’s friendships become more dyadic and intimate [45,46], and serve as a context for further learning about emotional experiences, norms for expression, and emotion regulation strategies. The unique nature of friendships may require adolescents to implement different emotion regulation strategies than those used in the context of parent–child relationships. Specifically, the fact that peer relationships become the primary source of social support during adolescence, surpassing parents in that role [43,44], suggests that adolescents may be less likely to use suppression and more likely to use affective expression as a form of emotion regulation. That is, adolescents may be more willing to express their emotions with friends in the belief and expectation that those emotions will be supported. Similarly, the motivation to maintain friendships may mean that relative to parent–child relationships adolescents are more inclined to give their friends the benefit of the doubt when it comes to evaluating negative relationship events by reappraising the friend’s behavior in a more positive light as a strategy to regulate their own emotional reaction to the event so as to maintain the friendship.

A majority of the empirical studies examining the role of peers in adolescent emotion socialization focuses on same-sex friendships and shows gender differences in emotion related processes in relationships with friends [47]. For example, Buhrmester and Furman [48] contend that boys exhibit intimacy in friendships through behavior, such as acts of assistance, rather than emotional disclosures and conversation. Consistent with this view, evidence indicates that females are more likely than males to express and discuss their emotions with friends [49,50,51]. Females’ friendships are also characterized as being more emotionally supportively than male friendships, with male friendships characterized by teasing and humor in response to emotional topics [37,45,51]. These gender differences are consistent with females’ greater relational orientation, and greater expectation of positive outcomes for disclosing emotions, compared to males [52,53], which may lead females to rely less on suppression as an emotion regulation strategy and be more likely to express emotion in the context of friendships than males [51]. To date, no study to the best of my knowledge has examined gender differences in adolescents’ use of emotion regulation strategies with friends.

### 1.3. Summary

The purpose of this study was to examine differences in adolescents’ reported use of emotion regulation strategies with parents and friends. In relation to this goal, it was hypothesized that adolescents would report using (H1a) less affective suppression, (H1b) more affective expression, and (H1c) more cognitive reappraisal with friends than with parents. It was also hypothesized that adolescents would report using (H2a) more affective suppression and (H2b) less affective expression with fathers than with mothers. A second goal of this study was to examine connections between reported emotion regulation strategies with parents and friends in relation to adolescent internalizing and external behaviors. Although the EPM suggests there may be relationship context differences in the use of ER strategies, the model does not offer guidance as to how such differences may impact adjustment outcomes. Rather, the theory suggests that the use of particular strategies, regardless of context, are indicative of dysfunctional emotion regulation. In this vein, the EPM was used to formulate six hypotheses: (H3a) reported use of affective suppression as a regulation strategy would be associated with higher levels of internalizing problems and (H3b) higher levels of externalizing behavior, (H3c) reported use of affective expression would be association with lower levels of internalizing and (H3d) higher levels of externalizing behavior, and (H3e) reported cognitive reappraisal would be associated with lower levels of (H3e) internalizing and (H3f) externalizing problems.

## 2. Materials and Methods

### 2.1. Participants

Participants were recruited from three middle schools randomly selected from the pool of 8 public schools in a midsized town (approximately 88,423 inhabitants) in the northeast United States. The sixth-grade population within each school was well integrated with African American, European American, and Latino students. Sixth-grade populations ranged from 30% African American, 34% European American, and 27% Latino students at one school to 33% African American, 26% European American, and 37% Latino in another. Families were recruited using rosters provided by school administrators to send letters describing this study and requesting permission to contact families during the summer before the target child entered seventh grade. Of 324 families contacted, 237 returned their forms giving consent to be contacted. Those families were called and screened for eligibility based on ethnicity, both parents living in the home and child health. A decision was made to require that parents be married to reduce variation in the family experiences of adolescents in this study that might confound relationship differences in emotion regulation strategies. A total of 185 families (72% of those contacted who were eligible) agreed to participate in this study. The sample included 55 African American, 67 European American, and 63 Latino youth, with 91 girls and 94 boys. As a condition for participation all adolescents came from homes in which their biological parents were married, with length of marriage ranging from 11–32 years (*M* = 16 years). Total family income ranged from $21,800 to $132,200 (*M* = $48,250). Of the mothers, 21% had less than a high school education, 45% had received a high school diploma, and 34% had completed an associate or bachelor’s degree. Of the fas, 15% had less than a high school education, 48% had received a high school diploma, and 37% had completed an associate or bachelor’s degree.

Child and parent ages were not significantly different across ethnic groups, *F*(2, 182) = 0.08, *p* < 0.88, *F*(2, 182) = 1.21, *p* < 0.62, and *F*(2, 182) = 2.31, *p* < 0.46, for adolescent, mother, and father age, respectively. The three ethnic group were comparable on socioeconomic indicators with no difference for mother education, *F*(2, 182) = 1.69, *p* < 0.55, father education, *F*(2, 182) = 2.48, *p* < 0.25, or median family income, *F*(2, 182) = 3.02, *p* < 0.18.

### 2.2. Procedure

During the summer prior to children’s entry into seventh grade, parent–adolescent dyads were invited to a research laboratory located on the University campus. Following procedures approved by the university Institutional Review Board (Project number: STUDY00002481), at the visit parents provided consent for their own and their child’s participation in this study, and adolescents provided assent to their own participation. During the visit parents and child completed questionnaires in separate rooms, and as each dyad in the family was observed in an interaction task. Data from these interaction sessions are not used in the current study, and therefore they are not described further.

### 2.3. Measures

#### 2.3.1. Family Demographics

Parents provided information concerning family demographics by responding to 14 questions that include annual household income, parent age, parental education, and the number of children in their household. Education was measured according to the number of years of education completed. There was a significant positive correlation between mother and father education, *r* = 0.58, *p* = 0.001. Therefore, to reduce the number of variables used in analyses, mother and father scores were averaged to form a parent education score. Parents reported total household income and household size. Mother and father reports were averaged. Income-needs ratios were calculated by dividing the reported household income by the federally determined poverty threshold for the number of individuals living in the household. Income-needs ratios above 1 indicate that a family is able to provide for basic needs, whereas ratios below 1 indicate that the family is unable to do so.

#### 2.3.2. Adolescent Emotional Regulation with Mothers, Fathers, and Friends

The Emotion Regulation Questionnaire for children and adolescents (ERQ-CA; [54]) was used to assess ER strategies of Cognitive Reappraisal (6 items) and Affective Suppression (4 items). Ten items are rated on a 5-point Likert-type response scale (1 = strongly disagree, 2 = disagree, 3 = half and half, 4 = agree, 5 = strongly agree). Higher scores on each scale reflected more use of the particular ER strategy. Past studies have demonstrated that the ERQ-CA has high internal consistency (α = 0.79 for Reappraisal, 0.73 for Suppression), 3 month test–retest reliability (*r* = 0.69 for both scales), and display convergent and discriminant validity when used to obtain data from adolescents [54,55]. For the purpose of this study the ERQ–CA was adapted by creating three different versions, one referring to each of three specific social partners, (a) mothers, (b) fathers, and (c) friends. In addition, the wording of certain items was simplified. For example, in the affective suppression scale the item “I control my emotions by not expressing them” was reworded to “When I am with my mother, I control my feelings by not showing them.” In the cognitive reappraisal scale the item “I control my emotions by changing the way I think about the situation I’m in” was reworded to “When I am with my father, I control my emotions by changing the way I think about the situation I’m in”. In this study, internal reliabilities of the Affective Suppression scale was 0.74, 0.71, and 0.77, and internal reliabilities for the Cognitive Reappraisal scale was 0.77, 0.81, and 0.86, for mother, father, and friend versions, respectively.

#### 2.3.3. Emotional Expressivity

The Emotional Expressivity Scale (EES; [56]) is a 17-item measure of the degree to which an individual communicates their emotional experience to others, verbally and nonverbally. A sample item is, “I cannot hide how I am feeling.” The intention of this measure is to assess the expression of emotion, separate from the experience or the interpretation of that emotion. Items are rated on a 5-point scale ranging from 1 (never true) to 5 (always true). To better address identified hypotheses, the EES was revised to create three versions to assess emotional expressivity when the participant was with “very close friends”, with “your mother”, and with “your father”. Revision to the measure included revision to the instructions and to the individual items. For example, the version inquiring about friends instructed the participant to “answer the following questions based on how you show your emotions to very close friends”, and an item was “I cannot hide the way I am feeling from my close friends”, whereas the version inquiring about their mother instructed the participant to “answer the following questions based on how you show your emotions to your mother”, and an item was “I cannot hide the way I am feeling from my mother”. The three versions of the EES were presented separately to the participants. In this study, internal reliabilities of the modified measures were 0.79, 0.84, and 0.80, for mother, father, and friend versions, respectively.

#### 2.3.4. Child Behavior Check List

Parents completed the Child Behavior Check List (CBCL) [57], a widely used and psychometrically sound measure. The 118 items assess symptoms across a broad range of clinical significance (e.g., from shyness to suicide attempts). In the current investigation the two CBCL scales for Internalizing and Externalizing were used. T scores above 70 are considered clinically significant, and scores between 65 and 69 are considered borderline significant. The CBCL has been used with acceptable levels of reliability to measure behavior problems of children aged 4–16 years in a variety of cultural settings [2,55].

These subscales demonstrated adequate internal reliability (mother α = 0.69–0.80, father α = 0.72–0.83). There was a positive skew in individual scores, with standardized values (skewness statistic divided by its standard error) ranging from 4.18–8.21 for mothers and from 3.94–7.62 for fathers. A square root transformation was used to reduce the skew of each distribution (standardized values of transformed scores ranged from −1.91–1.56 for mothers and from −1.87–1.60 for fathers). Centered transformed scores were used in correlations and regressions; mean item scores were used in t-tests. Several studies have demonstrated the distinctiveness and test–retest reliability of these scales during the adolescent years, and their strong concurrent associations with clinical assessments [2,57].

### 2.4. Plan of Analysis

Statistical analyses were performed using SPSS version 24.0 (IBM SPSS, IBM Corp., Armonk, NY, USA). Preliminary exploratory analyses (e.g., scatterplot, Q–Q plot, and correlation matrix) were conducted to examine the presence of potential outliers in this study. The graphs and correlation matrix showed that the data fit the premises, and no extreme outliers were identified. Next, a descriptive statistical analysis was conducted. Third, analyses were conducted to assess associations between demographic data, emotion regulation strategies and internalizing and externalizing scores. Finally, multivariate regression analyses were used to explore the association between the emotion regulation strategies and internalizing and externalizing symptoms. The level of significances was set at *p* < 0.05 (one sided) for all statistical analyses.

## 3. Results

### 3.1. Preliminary Analyses

Means and standard deviations of study variables are shown in Table 1. As can be seen, cognitive reappraisal was the most frequent emotion regulation strategy used by adolescents in all three relationship types. To examine the hypothesis that adolescents use different emotion regulation strategies in different relationships, a multivariate analysis of covariance (MANCOVA) was computed using a 2 (Male × Female) × 3 (Mother–adolescent × Father–adolescent × Adolescent–Friend) between-participants design, with family income as a covariate. The results revealed a significant main effect for child sex, Wilks’s λ = 0.86, *F*(2, 183) = 4.54, *p* < 0.01, η^2^ = 0.14, and relationship type, Wilks’s λ = 0.90, *F*(2, 182) = 3.82, *p* < 0.05, η^2^ = 0.10. There was no significant interaction effect between sex and relationship type Wilks’s λ = 2.13, *F*(2, 180) = 0.94, *ns*. Follow-up analyses revealed that boys reported using more affective suppression (*M* = 0.48, *SD* = 0.23) than girls (*M* = 0.34, *SD* = 0.32), *F*(1, 183) = 22.70, *p* < 0.001, η^2^ = 0.15. In turn, girls reported using more affective expression (*M* = 0.15, *SD* = 0.17) than boys (*M* = 0.08, *SD* = 0.13), *F*(1, 183) = 13.51, *p* < 0.001, η^2^ = 0.15. There was no significant difference between boys and girls reported use of cognitive reappraisal. Adolescents reported using more affective suppression with fathers (*M* = 0.14, *SD* = 0.18) than with mothers (*M* = 0.07, *SD* = 0.09), or friends (*M* = 0.07, *SD* = 0.09), *F*(1, 183) = 10.26, *p* < 0.01, η^2^ = 0.09. Adolescents reported using more affective expression with friends (*M* = 0.45, *SD* = 0.24) than with mothers (*M* = 0.38, *SD* = 0.30), or fathers, (*M* = 0.07, *SD* = 0.09), *F*(1, 183) = 8.46, *p* < 0.01, η^2^ = 0.08. Adolescents reported using more cognitive reappraisal with friends (*M* = 0.45, *SD* = 0.24) than with mothers (*M* = 0.38, *SD* = 0.30), or fathers (*M* = 0.07, *SD* = 0.09), *F*(1, 183) = 8.46, *p* < 0.01, η^2^ = 0.08.

### 3.2. Primary Analysis

Hierarchical multiple regression analyses were used to test the main hypothesis concerning the associations between adolescents’ emotion regulation strategies in different relationships and internalizing and externalizing behavior. The data were checked for potential violations of assumptions required for regression analyses. Specifically, multicollinearity among the independent variables was checked by looking at the bivariate correlations, variance inflation factor (VIF) and tolerance statistics. It was found that no bivariate correlation was higher than 0.77, all VIF values were between 1 and 10, and all tolerance values were higher than 0.10, indicating that the data met acceptable parameters for no multicollinearity. In addition, assumptions of multivariate normality were examined using normal quantile–quantile (QQ) plots and residual plots, univariate histograms, simple scatterplots, and univariate QQ plots that revealed no evidence of violations of normality in the data.

Prior to analyses, predictor variables were centered by subtracting each score from the mean score of the variable. Preliminary analyses revealed statistically significant (*p* < 0.05) correlations between adolescent outcomes and adolescent sex, and family income. Therefore, to conduct stringent examination of relations between emotion regulation strategies and psychosocial adjustment, sex and family income were entered in the first step of the regression to control for their effect. Step 2 contained the three emotion regulation variables with mothers. Step 3 contained the three emotion regulation variables with fathers. Step 4 contained the three emotion regulation variables with friends. Two separate regression analyses were conducted for internalizing and externalizing behavior.

Order of entering the emotion regulation variables within particular relationships into the regression analyses was arbitrary. A series of regressions also were computed in which the adolescent emotion regulation with father variables were entered in Step 2, followed by emotion regulation with mothers in Step 3 and emotion regulation with friends at step 4. In addition, a series of regressions in which the adolescent emotion regulation with friend variables were entered in Step 2, followed by emotion regulation with mothers in Step 3 and emotion regulation with fathers in step 4. The pattern of significant associations were the same in each set of regressions; thus, order of entry of the emotion regulation within relationship type variables did not affect the results reported in the following paragraphs. Results with emotion regulation strategies with mothers entered first, followed by emotion regulation strategies with fathers and emotion regulation strategies with friends are reported to maintain consistency with the order of information presented in other parts of this article.

#### 3.2.1. Predicting Adolescent Internalizing Behavior

As shown in Table 2, the predictors in the regression for adolescent internalizing behavior accounted for 22% of the total variance. After controlling for adolescent sex and family income, adolescent emotion regulation strategies with mothers and fathers were not significantly related to internalizing behavior. Adolescent emotion regulation strategies with friends accounted for a significant 7% of the variance in adolescent internalizing problems. Beta weights revealed that self-reported suppression of emotion with best friends was significant positively associated with internalizing behavior problems, and that expression of emotion with best friends was significantly negatively associated with internalizing behavior problems.

#### 3.2.2. Predicting Adolescent Externalizing

Additionally, as shown in Table 2, the predictors accounted for 33% of the variance in adolescent externalizing symptoms. After controlling for adolescent sex and family income, adolescent self-reported emotion regulation strategies with mothers were significantly associated with externalizing behavior problems. Beta weights revealed that both self-reported suppression of emotion and cognitive reappraisal were significantly negatively associated with externalizing behavior problems. Adolescent self-reported emotion regulation strategies with fathers also were significantly associated with externalizing behavior problems. Beta weights revealed that only self-reported cognitive reappraisal with fathers was significant negatively associated with externalizing behavior problems. Adolescent self-reported emotion regulation strategies with friends were not significantly associated with externalizing behavior problems.

## 4. Discussion

There is a dearth of research on adolescents’ use of emotion regulation strategies in different relationship contexts despite theoretical arguments that individual’s vary how they regulate emotion across different social contexts [26,27,28]. Empirical evidence demonstrates that emotion regulation in adolescence is associated with internalizing and externalizing symptoms [13,14], but it is necessary to examine how emotion regulation in different relationship contexts relate to problem behaviors to best understand the interpersonal processes associated with adolescent psychopathology symptoms. The current study was a first step toward understanding emotion regulation processes in both parent–adolescent and adolescent–friend relationships and how emotion regulation in these two relationship contexts is related to adolescent internalizing and externalizing behaviors. The findings of this study represent the first empirical support, to the best of my knowledge, for theoretical arguments from the EPM concerning the importance of distinguishing between relationship context in understanding connections between emotion regulation and psychological adjustment during adolescence [22].

Differences were found in the emotion regulation strategies that adolescents reported using with mothers, fathers, and best friends. Specifically, consistent with hypotheses, adolescents reported elevated use of affective expression and cognitive reappraisal with friends than with mothers or fathers. Affective expression is considered to be a form of emotion regulation based on the outward communication of emotion as a strategy to address the internal dissonance resulting from the experience of emotion [16,19,56]. Affective expression is an active form of emotion regulation intended to function by communicating to social partners that their actions may need to be changed because are evoking an emotional response [16,22,24]. Cognitive reappraisal involves the mental interpretation of an event in an effort to change the emotions that one is experiencing in reaction to that event [19]. It may be that adolescents’ greater use of these two emotion regulation strategies with friends than with parents is reflective of a developmental trend seen in adolescence in which youth become more autonomous from their parents, and orient more toward relationships with peers [31,42]. Adolescents may be more motivated to regulate their emotions with friends so as to preserve relationships that are more fragile than the well-established relationships with parents [44,49]. In a related vein, the difference in emotion regulation strategy use across relationships may reflect the fact that adolescents have different goals in relationships with parents and friends that guide strategies used to regulate the expression of emotion. Use of these strategies may be tied to the building of intimacy with friends that is a priority during the adolescent years [48]. Additional longitudinal research that examines children’s use of emotion regulations strategies with parents and friends across middle childhood into adolescence is needed to examine potential developmental changes.

Partial support was found for the hypothesis that adolescents would use less affective suppression with friends than with parents in that adolescents reported more affective suppression with their fathers than with their friends. However, there was no difference in adolescents’ reported use of affective suppression with mothers and friends. Affective suppression is a response-focused strategy that comes relatively late in the emotion-generative process, and primarily modifies the behavioral aspect of the emotion response tendencies [20,22]. It has been suggested that the use of affect suppression is influenced by the response that an individual expects to receive from the expression of emotion. Specifically, individuals may be more likely to use suppression of emotion in situations in which they expect to receive little support or to have little effect on their partner’s behavior by expressing emotion [10,16]. Research on the socialization of emotion expression in the family indicates that fathers are less responsive to children’s display of emotion than are mothers [36,37]. Thus, this finding may reflect a pattern of emotion regulation on the part of adolescents that is based on gender-typed parental responses to emotional expression. It may also be the result of the underlying relationship history that adolescents have with mothers, fathers and friends, with adolescents having learned to suppress emotions when interacting with their fathers. It will be of interest for future research to examine adolescents’ gender-typed beliefs regarding emotional expression and how differences in emotions expressed with mothers and fathers may be related to emotion regulation strategies.

The results of this study are also consistent with the view that emotion regulation is linked to mental health [11], and may have an important role in the emergence of internalizing and externalizing problems in early adolescence. This study expands existing knowledge by suggesting that ER strategies in different relationships are differentially linked to internalizing and externalizing outcomes. Only self-reported emotion regulation with friends was significantly related to internalizing behavior. Specifically, affective suppression was positively related to internalizing symptoms, whereas affective expression was negatively related to internalizing symptoms. This indicates that adolescents who report using more affect expression and less affective suppression with their friends were rated by parents as having fewer internalizing symptoms. This finding may reflect the fact that the social context of friendship is closely tied to self-evaluative emotions, such as depression and anxiety in adolescence. It has been suggested that suppression is detrimental to well-being because it leads to negative self-perceptions and feelings of being inauthentic in one’s social relationships [21]. Such negative self-perceptions may contribute to internalizing problems such as depression and anxiety. Another possible explanation is that over regulation of the expression of emotion with friends may interfere with the formation of intimacy in friendships. Given that intimacy is a key component of friendship quality and that adolescents have a high need for intimacy, a lack of intimacy may contribute to elevated feelings of depression and anxiety that are dominant features of internalizing symptoms.

Adolescents’ emotion regulation strategies with mothers and fathers made independent contributions to parent rated externalizing symptoms, but followed the same pattern. Specifically, use of affective suppression and cognitive reappraisal with both mothers and fathers was negatively associated with externalizing symptoms. Theoretical arguments concerning emotion regulation strategies suggest that cognitive reappraisal contributes to more optimal psychosocial adjustment compared to expressive suppression [12,18,19]. To the extent that cognitive reappraisal assists in the downregulation of negative emotion an individual will experience and express more positive, and less negative, emotion, and have higher levels of personal well-being [20,23]. Elevated experience of positive emotion is considered to reduce the enactment of externalizing behavior problems. Cognitive reappraisal with mothers and fathers may also be related to other information processing domains such as attributions.

By contrast, the EPM suggests that affective suppression is linked to poor psychosocial functioning. The model posits that affection suppression requires the individual to manage emotion response tendencies as they continually arise in a way that consumes cognitive resources that could otherwise be used for successfully navigating social interactions in which the emotions occur [22,58] Yet, in the present study, affective suppression with both mothers and fathers was associated with lower externalizing behavior scores. The fact that suppression with friends was not related to externalizing symptoms suggests that this may be a relationship-specific effect. That is, adolescents who suppress emotion with parents may be characterized by a high level of emotional control that reduces the likelihood of acting out. Emotion suppression may also reflect low levels of negative interaction and higher parent–adolescent relationship quality that is conducive to low levels of externalizing behavior. It may also be that adolescents who are less prone to externalizing behavior are more likely to use affective suppression as an emotion regulation strategy when interacting with their parents. It will be important for future longitudinal studies to trace the trajectory of the relationship between affective suppression with parents and externalizing behavior across the adolescent years.

The results of the present study underscore the merit of examining person–environment interactions in adolescents’ social relationships. Consistent with Magnusson and Magnusson’s [59] person–environment interaction perspective, the associations between emotion regulation and psychosocial symptoms varied based on the relationship context in which emotion regulation occurred. Nevertheless, it is important to note that the mean differences observed in emotion regulation across relationship types do not discount correlations indicating that at the individual level, adolescents’ report similar use of emotion regulation strategies in different relationships. Broadly speaking, research suggests that adolescents’ emotion regulation may be influenced by how relationship partners have responded to their expression of emotion the past [50]. Specific evidence suggests that the foundation for adolescents’ formation of close relationships outside the family, including establishing friendships with specific peers, may be laid in the context of the parent–child relationship through a process of niche selection [44]. For this reason it is important not to over interpret the distinctions between adolescents’ relationships with friends and parents that were found in this study because such relationships mutually influence one another.

The results of the present study join with other evidence suggesting that an examination of specific strategies yields a more nuanced picture of the association between emotion regulation and psychopathology symptoms [12,14]. However, it should be noted that it is common for individuals to deploy behaviors that embody assortments of diverse strategies rather than a single isolated strategy. The data of the current study are reflective of this in that a significant portion of the sample reported relatively high scores on multiple emotion regulation strategies as indicated by the correlations. Several recent conceptual accounts suggest that adaptability in choice of emotion regulation strategies is a critical ability for well-being and that numerous forms of psychopathology can be described by a malfunction in flexible regulation choice [19,58]. Healthy adaptation requires flexibility in changing between regulation strategies in a manner that is responsive to differing situational demands. In future work, consideration should be given to how patterns of variation in emotion regulation strategy use across contexts relates to adolescents’ psychosocial development.

There are a number of other limitations to this study that should be kept in mind when interpreting the data. Although the surveys used to assess emotion regulation in this study have been validated in previous research, the adaptation to focus on emotion regulation in specific relationships is novel to this study. There is questionable validity in assessing emotion regulation by having a person imagine the presence of another person. It is also questionable to what degree adolescents were able to accurately distinguish their emotion regulation strategy use with different partners. Even though the order of questionnaires were varied across adolescents, the fact that the surveys were completed at one time may have limited adolescents’ ability to distinguish between relationships when responding to the surveys. It is also possible that the ER questionnaires may have tapped relationship quality more broadly, rather than specific emotion regulation strategies. It would have been valuable to examine how participants’ parents perceived their children’s use of emotion regulation strategies and to examine potential discrepancies between adolescents’ and parents’ reports for emotion regulation.

The present study focused only on adolescents’ relationships with parents and friends, and on a narrow range of emotion regulation strategies. Future research should examine the ways in which aspects of each relationship type might affect the association between emotion regulation strategies and psychosocial adjustment. The present study examined whether this association differed in parent–adolescent and adolescent–friend relationships. However, adolescent marks a periods of expanding social relationships that may have implications for adolescent psychosocial adjustment. Future studies should consider adolescents’ emotion regulation strategy use with siblings and with romantic partners to further elucidate how relationship context may influence adolescents’ emotion regulation. In addition, future studies should examine additional forms of emotion regulation within different relationships.

This study did not address what processes may underlie the association between emotion regulation and psychosocial adjustment. The quality of the relationship between parents and adolescents and adolescents and friends could affect the degree to which emotion regulation relates to adolescents’ adjustment. The specific quality of a given relationship may play a moderating role in connections between the use of a particular emotion regulation strategy and adjustment outcomes. Likewise, relationship quality may account for individual differences in emotion regulation strategies within a given relationship context. Future research should examine the potential moderating role of relationship quality on connections between emotion regulation and adolescent adjustment outcomes. Future research that incorporates measurements of relationship quality and investigates diverse linkages between emotion regulation to psychosocial adjustment will build on the current study and lead to progress in understanding the ways in which emotion regulation may affect adolescent mental health.

## 5. Conclusions

In spite of these limitations, taken together, the findings suggest that relationship context plays a role in adolescents’ implementation of emotion regulation strategies. The results provide new evidence on the ways that emotion regulation strategies differ across relationships, and how relationship-specific emotion regulation strategies are related to psychosocial adjustment in adolescence. The findings suggest that differences in adolescents’ emotion regulation strategy use across relationships with parents and friends may be important in identifying distinct pathways to psychopathology. Self-reported emotion regulation with best friends was more robustly associated with internalizing symptoms than emotion regulation with parents. This may reflect the fact that peer relationships, friendship in particular, are linked to self-evaluations in adolescence. Adolescents’ report of emotion regulation with mothers and fathers was more significantly related to externalizing symptoms than was emotion regulation with friends. The longer history of the parent–child relationship may make patterns of emotion regulation that emerge in that relationship more meaningfully related to patterns of aggression and acting out that are manifestations of externalizing symptoms. To my knowledge, no other study has investigated how emotion regulation strategies in different relationships relate to adolescents’ psychosocial symptoms. The findings add to a growing body of evidence suggesting that emotion regulation is an important target of intervention. The findings that contextual circumstances amplify or alleviate emotion regulation risks have significant implications for the design of prevention and intervention programs for adolescents with internalizing and externalizing symptoms. If emotion regulation strategies can reduce behavior problems, developing strategies to teach adolescents emotion regulation strategies can reduce the risk of such problems.

## Figures and Tables

**Table 1 children-08-00299-t001:** Descriptive Statistics for adolescent self-reported emotion regulation with mothers, fathers and friends.

	Mother		Father		Friend			
	Boys	Girls	Boys	Girls	Boys	Girls	Gender	Relation
	*M (SD)*	*M (SD)*	*M (SD)*	*M (SD)*	*M (SD)*	*M (SD)*	*F*	*F*
Emotion regulation								
Suppression	14.21 (5.25)	13.16 (4.21)	14.67 (5.24)	13.53 (5.18)	14.20 (4.26)	13.14 (4.21)	11.26 **	5.82 *
Expression	57.25 (9.22)	59.30 (8.29)	55.20 (9.20)	58.27 (8.22)	62.28 (8.18)	66.37 (8.24)	9.25 *	6.08 *
Reappraisal	26.30 (6.30)	27.32 (6.28)	27.12 (6.23)	28.34 (6.25)	30.38 (5.22)	31.40 (5.20)	2.37	4.15 *
Internalizing	0.29 (0.31)	0.36 (0.23)	0.26 (0.27)	0.39 (0.25)			10.14 **	1.73
Externalizing	0.21 (0.22)	0.17 (0.27)	0.24 (0.23)	0.15 (0.25)			13.24 **	2.05

* *p* < 0.05, ** *p* < 0.01.

**Table 2 children-08-00299-t002:** Regression analysis: Relations of demographic characteristics, emotion regulation with mothers, emotion regulation with fathers, and emotional regulation with friends to adolescents’ internalizing and externalizing behaviors.

	Internalizing				Externalizing			
	Δ*R*^2^	*B*	*SE B*	β	Δ*R*^2^	*B*	*SE B*	β
Step 1: Demographics	0.12 **				0.15 **			
Adolescent sex		0.32	0.11	0.29 **		0.44	0.08	0.34 **
Family income		0.24	0.15	0.20 **		0.27	0.12	0.21 *
*F* for step	*F*(1, 183) = 5.40 **				*F*(1, 183) = 7.25 **			
Step 2: ER—Mother	0.02				0.06 *		
Affective suppression		1.12	0.81	0.16		−2.36	1.34	−0.36 **
Affective expression		−0.62	0.48	−0.10		0.72	0.24	0.16
Cognitive reappraisal		−0.84	0.76	−0.13		−1.71	0.81	−0.24 **
*F* for step	*F*(1, 180) = 0.40				*F*(1, 180) = 5.40 **			
Step 3: ER—Father	0.01				0.08 **			
Affective suppression		1.10	1.18	0.10		−2.22	1.34	−0.31 **
Affective expression		−1.12	1.88	−0.08		0.12	0.31	0.06
Cognitive reappraisal		0.64	0.04	0.03		−1.89	0.91	−0.27 **
*F* for step	*F*(3, 177) = 0.81				*F*(3, 177) = 4.72 **			
Step 4: ER—Friend	0.07 ^*^				0.04			
Affective suppression		1.83	0.87	0.44 *		0.35	0.28	0.17
Affective expression		−1.31	0.55	−0.30 *		0.15	0.19	0.08
Cognitive reappraisal		−1.08	0.91	−0.10		−1.92	1.12	−0.22
*F* for step	*F*(5, 174) = 4.52 *				*F*(5, 174) = 2.76			

Note: ER = emotion regulation. * *p* < 0.05, ** *p* < 0.01.

## Data Availability

Data files supporting reported results are available from the author upon request.

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
