# Peer review of "Emotion Regulation with Parents and Friends and Adolescent Internalizing and Externalizing Behavior"

_children, 2021, doi:10.3390/children8040299_

Round 1

Reviewer 1 Report

The authors have clarified the concerns raised in my previous review. I recommend the article for publication in the present form.

Reviewer 2 Report

I do not have further comments. The author has sufficiently addressed my concerns. 

This manuscript is a resubmission of an earlier submission. The following is a list of the peer review reports and author responses from that submission.

Round 1

Reviewer 1 Report

Eric Lindsey discusses the adolescent’s self-reported use of emotion regulation strategies with parents and friends in relation to internalizing and externalizing behaviors. The results provide new evidence on the ways that emotion regulation strategies differ across relationships, and how relationship specific emotion regulation strategies may be linked with the child’s internalizing or externalizing problems reported by the parents. The findings suggest that differences in adolescent’s emotion regulation strategies used across relationships with parents and friends may be relevant in identifying distinct pathways to psychopathology. Self-reported emotion regulation with best friend was more robustly associated with internalizing symptoms possibly reflecting the importance of peer relationships to self-evaluations in adolescence. Adolescent’s report of emotion regulation with mother and father was more significantly related to externalizing symptoms, both self-reported suppression of emotion and cognitive reappraisal were significantly associated with externalizing behavior problems, the relations being negative.

The main contribution of the paper is that the relationships with parents and friends are examined as two distinct contexts in which adolescents may use different emotion regulation strategies. In plus, the author examined the differences in adolescents’ emotion regulation strategy use with mothers and fathers. The internalizing and externalizing problems were assessed by both parents which is again a plus.

The title and abstract are appropriate for the content of the text. Furthermore, the article is well constructed, the design and analysis were well performed, and the result are clearly presented. The conclusions drawn are adequately supported by the results.

Minor comments:

Page 3; line 139 paragraph “For example, [34] reported that mother’s negative expressivity was related to higher depressive symptoms in adolescents through adolescent emotion dysregulation”, the name of the author should be added.

Three emotional regulation strategies have been found in studies to be protective for the emergence of psychopathology: reappraisal, problem-solving, and acceptance. Suppression (emotions and thoughts), avoidance (experiential and behavioural), and rumination are considered risk factors. Emotional expression is less studied in clinical populations. Age moderates the relationship between psychopathology and some non-adaptive emotional regulation strategies (i.e., suppression). Children are less able to use them due to the lack of acquisition of executive control over emotional reactions or difficulty in using metacognition. Appropriate emotional regulation is a critical component for optimal child function, but there are differences in the use of emotional regulation strategies depending on the developmental level and the presence of a clinical disorder. There is some research regarding the link between emotion regulation strategies and the externalizing problems, suggesting that emotion regulation using avoidance and suppression may increase aggression by exaggerating negative emotions, reducing aggression inhibition, compromising decision-making, diminishing social networks, increasing physical excitement, and preventing solving difficult situations. Deficits in emotion regulation strategies directly (i.e., acting aggressively due to intense anger) or indirectly (i.e., by making the child more difficult to discipline) lead to the development of externalizing problems. Other instruments designed to assess emotional regulation strategies, classifies a number of these strategies as adaptive (i.e., positive refocusing, cognitive reappraisal, putting in perspective, refocus on planning, and acceptance), and several others as maladaptive (i.e., rumination, self-blame, blaming others, and catastrophizing). Among the studies that have utilized this measure, different strategies have been differentially related to outcomes regarding the psychopathology. Overall, the important point, which has been emphasized throughout this article, is that these strategies are unlikely to be universally adaptive or maladaptive, but situationally dependent.

Reviewer 2 Report

I enjoyed reading this article. It focuses on the different emotion regulation strategies adolescents use with their parents and friends and the relations between the use of these strategies to internalising and externalising problems. I think that it is an important next step to zoom in on the emotion regulation strategies of adolescents in their different relations. I have a number of smaller and some bigger concerns.

Introduction.

Line 68. It is mentioned that there are five types of emotion regulation strategies. I then expected that these five strategies would be introduced. However, only two types or subtypes are described and used in the study. I suggest rephrasing in order not to confuse the reader.

Moreover, in the hypothesis emotional expressivity is mentioned. In the discussion this is described as a third ER strategy, but it is not introduced as such in the introduction, which should be added. I am not convinced that emotional expressivity is correctly described as an ER strategy, as it measures the outcome of the emotion process (thus the expression of an emotion after emotion regulation).

Line 197. It is hypothesised that adolescents use less affective suppression with friends than with parents. However, on line 172 it is argued that children are likely to express emotions to a lesser extent to friends in order not to endanger their friendship. This seems contradicting. What was the hypothesis based on?

Throughout the introduction I was wondering whether the focus was on emotion regulation within the relationships, or emotion regulation strategies socialised through the relationships. For instance, when female friendships are more emotionally supportive, this is probably measured by asking whether emotions in general are shared and discussed (so a tendency to discuss emotional experiences is socialised/ used in these relationships). It is another topic whether girls express/suppress their emotions more towards each other in their relationships. This article seems to focus on the way emotions within the relationships are expressed towards each other and not which strategies are socialised in general. Can both be distinguished in the used methods? Some clarification of the focus of the study would be helpful.

Methods

Why was marriage a prerequisite for participating in the study? Does it not create unnecessary bias to exclude parents who live together without being married?

Line 271 difference = different

Line 262; it would be informative to add an example item to the appraisal scale.

Results

Are there differences in ER strategies in the cultural groups?  As ER strategies are socialised, this seems likely and interesting to examine.

Table 1; please add the range of the scales.

I am having doubts about the regression analyses. Why is chosen for a step-wise regression? And how is the order in which ER strategies with mothers, fathers and friends are added to the model chosen? Although there is no indication of multicollinearity, it is likely that there is shared variance obscuring possible relations. I would expect there to be correlations between the strategies used with different relationships. A correlation table between the ER strategies with mothers, fathers and friends and with the internalising and externalising problems would be helpful to understand the examined relationships. Additionally, entering the predictor variables backwards might be better when there is no theoretical logical order of entering the variables. 

Based on the gender effects discussed in the introduction, it would be interesting to examine whether there are interaction effects in the relations between ER strategies and internalising and externalising problems for boys and girls separately.

Discussion

Future studies may also want to include multiple different ER strategies in addition to the strategies examined in the current study.

Are there implications for interventions of differences between ER strategies with friends and parents?